# Health-Promoting Properties of Medicinal Mushrooms and Their Bioactive Compounds for the COVID-19 Era—An Appraisal: Do the Pro-Health Claims Measure Up?

**DOI:** 10.3390/molecules27072302

**Published:** 2022-04-01

**Authors:** Jennifer Mary Phillips, Soo Liang Ooi, Sok Cheon Pak

**Affiliations:** 1School of Dentistry and Medical Sciences, Charles Sturt University, Bathurst, NSW 2795, Australia; jennifer-phillips@live.com.au (J.M.P.); sooi@csu.edu.au (S.L.O.); 2LAGOM NutriHealing, 16 Gentile Court, Hobart, TAS 7010, Australia

**Keywords:** COVID-19, *β*-glucans, immunomodulation, anti-inflammation, anti-oxidant, ACE2 regulation

## Abstract

Many mushroom species are consumed as food, while significant numbers are also utilised medicinally. Mushrooms are rich in nutrients and bioactive compounds. A growing body of in vitro, in vivo, and human research has revealed their therapeutic potentials, which include such properties as anti-pathogenic, antioxidant, anti-inflammatory, immunomodulatory, gut microbiota enhancement, and angiotensin-converting enzyme 2 specificity. The uses of medicinal mushrooms (MMs) as extracts in nutraceuticals and other functional food and health products are burgeoning. COVID-19 presents an opportunity to consider how, and if, specific MM compounds might be utilised therapeutically to mitigate associated risk factors, reduce disease severity, and support recovery. As vaccines become a mainstay, MMs may have the potential as an adjunct therapy to enhance immunity. In the context of COVID-19, this review explores current research about MMs to identify the key properties claimed to confer health benefits. Considered also are barriers or limitations that may impact general recommendations on MMs as therapy. It is contended that the extraction method used to isolate bioactive compounds must be a primary consideration for efficacious targeting of physiological endpoints. Mushrooms commonly available for culinary use and obtainable as a dietary supplement for medicinal purposes are included in this review. Specific properties related to these mushrooms have been considered due to their potential protective and mediating effects on human exposure to the SARS CoV-2 virus and the ensuing COVID-19 disease processes.

## 1. Introduction

Mushrooms have long been regarded as healthful and widely consumed for their culinary and nutritional values. Some species have an ancient tradition as medicinal therapies, and increasingly, this is being realised in a contemporary context [1,2]. However, until recently, the scientific understanding of mushrooms’ application as a medicinal agent has been chiefly empiric [3]. Interest in advancing the health properties and pharmacological activities means that clinical research is growing, and much of the traditional knowledge is being documented and validated [4]. Indeed, there is an interdisciplinary field of science studying medicinal mushrooms (MMs), with an increasing number of human studies emerging. This, along with industry technological developments, means that some mushrooms are now regarded as a class of drugs called “mushroom pharmaceuticals” [1]. Physiological activities revealed from numerous studies include anti-pathogenic, antioxidant, anti-inflammatory, immunomodulatory, and anticoagulation effects, plus gut microbiota enhancement, pulmonary cytoprotection, and angiotensin-converting enzyme (ACE) 2 specificity [4,5]. These actions are mainly attributable to the bioactive compounds present in the fruiting bodies and the mycelium, depending on the species [2]. For example, *Ganoderma lucidum* contains more than 120 different triterpenes plus polysaccharides, proteins, and other bioactive compounds [6].

Coronavirus disease 2019 (COVID-19), caused by the severe acute respiratory syndrome coronavirus 2 (SARS CoV-2), presents an opportunity to consider how macro-fungi or their derivatives might be harnessed and utilised as therapeutic agents. In particular, this pertains to their use in optimising health to avoid or mitigate risk factors, prevent severe disease outcomes, and improve recovery prospects as commonly noted or experienced. For example, the immune response is crucial in COVID-19 pathogenesis, and its modulation to control the hyperinflammatory response would be advantageous. Additionally, the virus appears to have a higher prevalence for and more severe outcomes in populations with particular risk factors and comorbidities associated with age, obesity, metabolic, cardiovascular, and inflammation-mediated conditions [7,8]. Many of these factors are known to respond to, and are modifiable through, dietary or lifestyle interventions. Therefore, with their potential and specific therapeutic effects, MMs seem well placed to be considered as nutraceutical options, or at the very least, as health-promoting food sources. Moreover, as vaccines become the mainstay for preventing severe outcomes, finding effective immune-enhancing adjuncts will be beneficial.

MMs are increasingly being utilised as a nutritional food source in the “food as medicine” health-promoting dietary approach [9]. They are also used as dietary supplements, biocontrol agents, and cosmetics. Most pertinent is their use as nutraceuticals and natural products for pharmacological therapy, which are valued for their immunological, anti-inflammatory, and health-promoting effects [4]. However, of prime consideration is that mushrooms’ properties, mechanisms of action, and potential efficacies can be influenced by many variables, including climate, location, cultivation, processing, and extraction techniques [4,10]. The nomenclature may also be a problem as it affects accurate identification and, thus, attribution. As more research is achieved, and commercialisation and application continue to increase, these variables must be addressed. However, more work may yet be required. Particular aspects such as extraction methods may be very pertinent in this quest.

In the broader context of COVID-19, this narrative review investigates the use of mushrooms as medicine, exploring the bioactive compounds they contain and the associated pro-health claims. Specific properties have been considered due to their potential health enhancement or mediating effects on human exposure to the SARS CoV-2 virus or the ensuing COVID-19 disease processes. Mechanisms of action, physiological effects, and potential capabilities such as antiviral, antioxidant, immunomodulatory, cardiovascular regulation, and health-promoting factors will be highlighted.

The research question has two parts; (1) does the research support the potential of MMs to enhance health and protect against or ameliorate symptoms associated with COVID-19, and (2) what limitations currently impact utilising mushrooms as a reliable medicinal treatment? Research findings associated with specific mushrooms such as *Agaricus* spp. (e.g., *bisporus*, (common white button/brown)), *Cordyceps militaris*, *Flammulina velutipes/enokitake* (Enoki), *G. lucidum* (Reishi), *Grifola frondosa* (Maitake), *Hericium erinaceus* (Lion’s Mane), *Lentinus edodes* (Shiitake), *Pleurotus* spp. (e.g., *ostreatus*, (Oyster)), and *Trametes versicolor* (Turkey Tail) are considered.

## 2. Research Methods

Major databases were initially searched via the Primo search tool of Charles Sturt University Library using the keywords “mushrooms” and “health” and “COVID-19” or SARS CoV-2”. Further searches included specific mushroom species such as “*Ganoderma* spp.” and compounds such as “β-glucans” and health effects such as “inflammation” or “immunomodulation”. Reference lists associated with pertinent papers were examined extensively for additional source material.

## 3. SARS CoV-2 Virus and COVID-19 Infection

Human coronaviruses such as the common cold generally cause mild illnesses but can mutate over time [11]. SARS CoV-2, a novel species of the Coronaviridae family, is the infective agent that causes COVID-19. Structural elements of the virus critically enable attachment, cell activity, multiplication, and infectiousness [12], with ACE2 being the primary entry receptor [13]. SARS CoV-2 has become a global concern due to its mutative nature, transmissibility, virulence factors, and the severity of the disease processes [12]. SARS CoV-2 enters the lungs through the respiratory tract, directly infecting upper and lower tract cells. Infection typically involves the upper respiratory area and nasal ciliated epithelial cells, the lung, and alveolar epithelial cells [7,13]. Additionally involved are other types of endothelial cells of the arteries, immune, smooth muscle, and intestine [7]. This presents implications for systemic organ involvement, particularly through the pro-inflammatory immune response that characterises the disease sequelae.

Presentation of the disease can range from no symptoms (asymptomatic) to severe illness with potentially life-threatening complications or fatal outcomes [11]. For the vast majority, symptoms are similar to those of mild to moderate influenza. Present may be fever, dry cough, sore throat, muscle pain, and fatigue in the initial phase, and it can extend to headache, anorexia, malaise, dyspnoea, nasal congestion, haemoptysis, diarrhoea, lymphopaenia, and difficulty with reading and distinguishing smells in the ensuing phases [7,12]. Continuing respiratory distress can lead to acute respiratory distress syndrome (ARDS), requiring intensive medical intervention [12]. While most people recover, at increased risk for severity are the immunocompromised, elderly, male gender, and people with comorbid conditions such as cardiovascular diseases (CVD), diabetes, hypertension, and poor nutritional status [7,14,15].

## 4. Immune and Inflammatory Responses to COVID-19

Host innate and adaptive immune cells, particularly lymphocytes (T cells, B cells, and natural killer (NK) cells), are called upon to defend against SARS CoV-2 viral invasion and tissue damage [12,15]. The optimal immune response in humans involves the coordination of cytokine and chemokine activation, recruitment of defence cells, and secretion of antibodies in a timely, localised, and functional way. The combined immune and antiviral activation is tightly regulated and balanced to eliminate and resolve the viral invasion and promote tissue repair [15].

However, interference with, and aberration of, the immune responses along with associated hyperinflammation is a key characteristic of the increased disease severity of COVID-19 [12,13]. In the progression from a normal response to one that may lead to death, total T cells along with CD8+ and CD4+ required for clearing viral invasion are known to decrease markedly, leading to cell exhaustion and dysfunction [14,15,16]. Triggered, recruited, and increased in a highly organised cellular and molecular cascade are pro-inflammatory type 1 and type 2 cells, particularly interleukins of IL-6, IL-10, IL-2, and IL-7, as well as granulocyte, colony-stimulating factor, and tumour necrosis factor (TNF) [12,13]. There is also interference in the native immune response generated via Toll-like and other receptors. This triggers the expression of interferons and activates antiviral effectors such as NK cells and macrophages, and DNA replication and transcription anomalies triggering apoptotic pathways [7].

Systemic manifestations involve severe inflammation, respiratory complications, alterations of the circulatory system through endothelial cell interactions and damage to the vascular barrier, capillaries, and organs, dysregulation of the ACE2 receptor gateway, and disruption of the renin–angiotensin–aldosterone system (RAAS) [14,15,17]. Furthermore, cardiometabolic disease risk factors are implicated in the increased severity, morbidity, and mortality associated with COVID-19. Of note is that cardiovascular mortality is greater in all influenza pandemics than in all other causes, and acute respiratory viral infections are triggering factors for cardiovascular disease. Events such as myocardial infarction and inflammation, thromboembolism, and vasculitis are evident, and high blood pressure, obesity, and diabetes are known comorbidities [7,18,19].

## 5. Mushrooms as Prevention or Treatment for COVID-19

### 5.1. General Features

Mushrooms are macro-fungi with a distinctive fruiting body, either hypogeous (underground) or epigeous (aboveground). There are estimated to be around 140,000 known species of macro-fungi belonging mainly to the phyla Basidiomycetes and some to Ascomycetes. Many more hundreds of thousands of fungi are assumed yet to be identified and classified [4,6,20]. Some 2000 are edible, and a few hundred wild and cultivated mushrooms have long been utilised as MMs [5,20]. Nutritionally, mushrooms are low in energy and are generally a good source of macro and micronutrients and trace elements, although there is variability [21,22,23,24]. As functional foods, they are also anti-inflammatory and known to modulate gut bacteria [25]. Species-specific structural and maturation elements of fruiting bodies and mycelia, and potentially even the fermented substrate from which the prepared product may have been produced, impact effects. Variances involve the chemistry, bioactive fractions derived from them, and biological activity [2,17,26].

### 5.2. Structural Elements and Bioactive Compounds

MMs produce bioactive primary and secondary metabolites and specific molecular weight compounds such as polysaccharides, polysaccharide–protein complexes, polyphenols, terpenoids, lectins, coumarins, ribosomal and non-ribosomal peptides, peptidoglycans, alkaloids, fatty acids, sterols, and antioxidants [1,2,5,6,20,27,28,29]. The most studied health-promoting properties and effects of MMs appear to be related to polysaccharides, lectins, protein complexes, sterols, and polyphenols such as terpenoids [2,17]. For example, lectins are non-immunoglobulin binding storage proteins that play a crucial role in such biological processes as cell signalling, cell–cell interactions in the immune system, and host defence mechanisms [20]. Mushroom polysaccharides have a strong ability to carry biological information, and via such function, they have antitumour, antioxidant, immunomodulatory, anti-inflammatory, antimicrobial, anti-obesity, and anti-diabetic effects [18]. To some extent, all of the bioactive compounds mentioned above have a body of knowledge that could be examined in respect of SARS CoV-2 virus protection and COVID-19 symptom reduction, as these compounds demonstrated beneficial effects that may be very helpful. Notably, these effects include inflammation regulation, immune and reactive oxygen species (ROS) modulation, host defence mechanisms involvement, microbial activity, pulmonary cytoprotection, and ACE2 specificity or inhibition [4,17,30].

However, β-glucans (D fraction) appear to have wide-ranging effects and benefits that deserves further examination. Additionally, mushroom-derived β-glucans are becoming more common in nutraceutical products and are promoted in a functional culinary sense. The limitations that may impact the appropriate application must also be considered. For example, their efficacy is very dependent on species, extraction techniques, and methods.

### 5.3. β-Glucans

Glucans are heterogeneous polysaccharides, with short or long-chain glucose polymers linked by large numbers of glucose sub-units, different branching, branch linkage, and backbone structures. They are a natural component of the cell walls. In addition to mushrooms, glucans are also a constituent of foods such as cereals. Thus, to be clear, the fungi glucans, in particular, β-glucans, are discussed and referred to as edible mushroom polysaccharides (EMPs).

The molecular weight, chain length, and side-branching structures of mushroom β-glucans can be species and cultivar specific, influencing variance, complexity, and biological activity [5,17,31,32]. The purity and purification processes are essential in characterising the structure [25]. Growing environments, drying conditions, and isolation/extraction methods are also crucial for the intensity of β-glucans’ activities [18]. The solubility and particulate size of β-glucans are important physical features as many receptors involved can activate different immune responses [31]. For example, particulate β-glucans are known to directly stimulate immune cell activation through a dectin-1 pathway, while soluble β-glucans require complement receptor-3 dependent pathway activation.

The immunomodulatory properties of mushroom-derived β-glucans exert more potent immunoinflammatory effects than other types and have long been recognised for this [31,33]. Immunomodulation is characterised by the ability to correct deviated immune functions. This may be through supporting declined or suppressed parameters or normalising overactive or increased functions [34]. Notably, due to their confirmed complex mode of action, β-glucans are recognised as biological response modifiers (BRMs). They induce epigenetic programming in innate immune cells to produce a more robust immune response and act as pathogen-associated molecular patterns, binding to specific pathogen recognition receptors, inducing innate and adaptive immune responses [31,33]. They can also stimulate the activity of macrophages [35] and neutrophils, support NK cell activity, influence the production of cytokines and chemokines, and modulate antibody production, amongst many other functions [34].

Several authors have described the concept of trained immunity [14,33,36] as being a modified and epigenetic innate immune response capable of producing antibody-free memory to a secondary heterologous stimulus that is more robust. However, Geller and Yan [36] cautioned that due to the development of hyperinflammatory symptoms, such as those that may occur in COVID-19, the use of a therapy that could induce trained immunity effects warrants particular consideration. Notwithstanding, due to their immune enhancement benefits, studies examined the use of biomacromolecule compounds, including polymers such as β-glucans and chitosan as vaccine adjuvants [37,38,39]. Moreover, the potential of using β-glucans as a wide-spectrum immune-balancing food-supplement-based enteric vaccine adjuvant for COVID-19 was explored by Ikewaki et al. [33].

## 6. Systemic Pro-Health Responses and Activities Associated with Specific MMs

The following synopsis examines the various pro-health effects and activities of specific macro-fungi elucidated in research that may have applications to the pathological sequelae and outcomes associated with COVID-19. The effects and potential benefits of MMs are illustrated in Figure 1.

### 6.1. Anti-Pathogenic

The basic viral cycle, also associated with SARS CoV-2, involves attachment, penetration, uncoating, replication, assembly, and release [40]. Mushroom extracts and bioactive compounds impede viral entry into host cells and multiplication, inhibit virus adsorption, replication, nucleic acid synthesis, and disrupt other pathogens [3,6,14]. It is known that proteolytic enzymes facilitate the cleavage of S glycoprotein, which is a critical step in SARS CoV-2 viral attachment [12]. Since protease inhibitors have been isolated from *G. lucidum*, *C. militaris*, and *A. bisporus* [14], MMs may have therapeutic utility.

Cordycepin isolated from *C. militaris* exerted an antiviral effect through a protein kinase inhibitory mechanism and an inhibitory role towards ribonucleic acid (RNA) synthesis and Epstein–Barr virus (EBV) replication [41]. Ganoderma compounds isolated from *G. lucidum* effectively inhibited human immunodeficiency virus (HIV)-1 and HIV-1 protease [42]. Additionally, various triterpenoids (isolated from *G. lucidum* and other Ganoderma species) were active against HIV-1, influenza type A, and herpes simplex type 1 [6]. In vitro and in vivo studies on a range of common viral agents from *Agaricus* spp. including *Agaricus blazei* Murril (AbM), *H. erinaceus*, and *G. frondosa* demonstrated antiviral properties [3].

MMs are also purported to have anti-bacterial functions [1]. To confirm this potential, Hearst et al. [43] conducted an in vitro microbiological assessment using aqueous extracts of *L. edodes* and *P. ostreatus*. The aqueous extract of *L. edodes* demonstrated potent activity when tested in culture against 29 bacterial isolates (Gram-positive and Gram-negative) and 10 fungal/yeast agents. Here, 85% of the bacterial and 50% of the fungal organisms were inhibited by the *L. edodes* extract. The results compared favourably against Ciproflaxin, which is a broad-spectrum antibiotic that was deployed as the control. In contrast, *P. ostreatus* aqueous extract showed minimal activity on the same range of pathogens, with only three out of 39 samples inhibited, while none of the yeast and mould species was affected. Additionally, a purified source of lentinan, a specific class of β-glucan, reduced populations of multiple antibiotic-resistant clinical isolate *Klebsiella pneumoniae* in an in vivo lung infection model and showed potential for treating sepsis-induced lung injury and boosting type 1 interferon response to RNA viruses such as influenza and coronavirus [17].

### 6.2. Immune Modulation

A response elicited from *L. edodes* named “the lentinan antiviral effect” has been attributed to innate immune responses and specific immunity regulation. Acting as a BRM, lentinan can promote T helper cell (Th) type 1 response and improve Th1/Th2 balance. It may also activate inflammasomes, enhance immune cells, activate the complement system, and promote cytotoxicity and phagocytosis [2,44]. An in-house hot water extract of *L. edodes* was compared to a commercially sourced lentinan extract (Carbosynth–Lentinan (CL)) to investigate if isolates could alleviate the immune cascade in conditions experienced by COVID-19 patients, such as ARDS. β-glucans from *L. edodes* reduced IL-1β and IL-6 in lung injury and activated macrophages in vitro [17]. β-glucans were also used to investigate oxidative stress alleviation in H_2_O_2_-treated THP-1 cells. Viability, apoptosis and necrosis were assessed. CL extract attenuated oxidative stress-induced early apoptosis, and the in-house lentinan extract attenuated late apoptosis [17].

Lectin derived from *P. ostreatus* has been studied as a hepatitis B virus DNA vaccine adjuvant and demonstrated effectiveness in enhancing surface protein antibodies [45]. Studies utilising pleuran (insoluble β-glucans derived from *P. ostreatus*) administered in oral liquid syrup form have suggested numerous positive immunomodulatory effects in recurrent upper and lower respiratory tract infection (RRTI). Demonstrated effects of pleuran, particularly in studies with children, include reduced incidences of RRTI, otitis media, tonsillopharyngitis, bronchitis, laryngitis, and other flu and cold-like symptoms, plus fewer days off school [46,47,48].

AbM extract is another rich source of BRMs. Via the actions of, for example, proteoglycans, β-glucans, and ergosterol, anti-inflammatory, anti-pathogenic, and immunomodulatory cytokine effects were stimulated, vaccine efficacy was improved, and cytotoxic effects were induced [49,50,51,52]. Andosan™, a product primarily manufactured from AbM extract, combined with *H. erinaceus* and *G. frondosa*, has been investigated in clinical studies [3]. Independently, these three mushrooms have demonstrated efficacy for their immunomodulatory, anti-infective, antitumour, and anti-inflammatory effects with reduced pro-inflammatory cytokines and oxidative stress, and beneficial gut microbiota responses [52]. *H. erinaceus* contains aromatic compounds such as hericerins and erinacines that appear to function as a nerve growth factor as well as the beneficial immunomodulating and antitumour properties derived from the glycoproteins and polysaccharides [52]. Further highlighting this, polysaccharides extracted from liquid-cultured mycelia and fruiting bodies of *G. frondosa* demonstrated antioxidant, antitumour, anti-inflammatory, hepatoprotection, and immunostimulatory activity [53]. Grifolan, a β-glucan isolated from *G. frondosa*, showed enhanced cellular immunity and modulation activities evidenced by increasing IL-2 and IL-10 production and augmentation of IL-6, IL-1, and TNF-α expression [54].

Water extract of four different MMs, including *G. lucidum*, caused NK cell-induced cytotoxicity against cancer cells, but an ethanol extract did the opposite by reducing intracellular pathway activation [55]. Various triterpene acids and sterols isolated from *G. lucidum* fruiting bodies revealed antitumour and anti-inflammatory effects as demonstrated via induction of EBV early antigen by 12-O-tetradecanoylphorbol-13-acetate [56].

*T. versicolor* has a long traditional history of use to promote health, strength, and longevity. More recently, numerous studies, including clinical trials, suggest properties and effects that include antimicrobial, antiviral, antitumour, anti-inflammatory, antioxidant, hepatoprotective, bone protective, and notably immunopotentiation [57,58]. Two bioactive mycelia extracts of protein bound polysaccharides from *T. versicolor*, namely polysaccharopeptide (PSP) and polysaccharide krestin (PSK), are currently utilised medicinally in some countries as integrated cancer therapy and adjuncts for chemotherapy and radiotherapy [2,57,59]. From a range of randomised and non-randomised controlled trials, both PSK and PSP promoted positive impacts on anticancer effects [60]. Deemed resulting from the immunomodulation and potentiation of immune surveillance, PSK and PSP positively affect immune parameters, haematological function, performance status, quality of life, body weight, fatigue, pain, nausea, anorexia, and median survival [59,60,61]. Additionally, antitumour and antimetastatic effects were noted through direct tumour-inhibiting experiments in vivo [60]. Of interest in the context of COVID-19 application is the mechanisms of *T. versicolor*. This appears to be through the inducement of predominantly pro-inflammatory cytokines: not only those associated with TNF-α and NK cells but also pleiotropic cytokines such as IL-1α and 1β and IL-6, plus prostaglandin E2, histamine, activation of complement-3, and T cell proliferation [57,59,61]. While this may be desired to improve cancer outcomes, such as enhancing the immunosuppressive status, a cautionary approach in applying *T. versicolor* due to the hyperinflammatory response associated with COVID-19 progression should be taken. However, perhaps, there may be a place for consideration in the context of long COVID or playing a role as a vaccine adjuvant.

### 6.3. Antioxidant

The antioxidant/ROS system plays a significant role in pathogenic protection, regulation, and homeostasis in the human body. For example, the increased activity of ROS is a key feature in the pathogenesis and progression of many disease states such as atherosclerosis, arterial thrombosis, hyperlipidaemia, hypertension, cancer, obesity, insulin resistance, diabetes mellitus, hepatic and renal conditions, amongst many others [62]. These disease states are representative comorbidities associated with SARS CoV-2 and COVID-19 sequalae and experience. The antioxidant capacity of MMs has been demonstrated in various studies through radical scavenging, lipid peroxidation inhibition, and increasing antioxidant enzyme activities [30,54,63,64]. Bioactive compounds such as phenolics, indoles, flavonoids, glycosides, polysaccharides, tocopherols, glutathione and ergothioneine, ascorbic acid, carotenoids, vitamin D, copper, manganese, zinc, and selenium in MMs all participate in reducing oxidative stress [62,65]. Ergothioneine deserves special mention, as it has a vast array of unique cytoprotective properties pertinent to COVID-19 pathologies, including scavenging reactive oxygen and nitrogen species. It is able to modulate inflammation, inhibit the expression of vascular adhesion proteins, and protect against respiratory burst, amongst many other antioxidant activities [63]. Notable amounts of bioavailable ergothioneine were demonstrated in the fruiting bodies of *A. bisporus* [66], *L. edodes*, *P. ostreatus*, and mycelia of *C. militaris* (strain cm5), *H. erinaceus*, and *P. eryngii* [67,68].

Liquid–liquid partitioned fractions of *H. erinaceus* were evaluated for their anti-atherosclerotic potential through evaluation of in vitro inhibitory effect on low-density lipoprotein (LDL) oxidation and 3-hydroxy-2methylglutaryl coenzyme A (HMG-CoA) reductase activity [69]. Several bioactive compounds with antioxidant activity were isolated, in particular ergosterol. Hexane solvent fraction demonstrated the most potent inhibiting oxidisation of LDL and HMG-CoA reductase activity. This indicates a possible role in preventing oxidative stress-mediated vascular disease processes [69].

Radical scavenging properties associated with catalase activity, glutathione reductase, and glutathione peroxidase activities were demonstrated in varying degrees from methanol and water extracts isolated from the gills, stipe, and caps of two wild strains and one cultivated strain of *A. bisporus* [70]. Fourteen selected culinary MMs were evaluated for in vitro antioxidant and ACE inhibitory activities [30]. The mushrooms were extracted by boiling water for 30 min. The total phenolic content was determined with *G. lucidum* demonstrating the highest phenolic content and the most potent ACE inhibitor. Antioxidant capacity was carried out via measuring the free radical scavenging effect, β-carotene, lipid peroxidation, reducing power ability, cupric-ion-reducing antioxidant capacity, and ACE inhibition. An antioxidant index was determined based on the average percentage relative to quercetin. *G. lucidum* and *H. erinaceus* were shown to be relatively high compared to the other mushrooms [30].

### 6.4. ACE2 Regulation

The deleterious effects of COVID-19, such as those associated with cardiometabolic and other hallmark disorders, demonstrate dysregulation of the homeostatic function within the RAAS [7,13,71,72]. RAAS maintains dynamic control of vascular function. ACE2 is an integral membrane protein present in the lungs, liver, heart, kidney, and endothelium. ACE2 dysregulation appears to strongly impact the RAAS, manifesting effects involving hyperinflammation and oxidative stress. MMs have been investigated for ACE inhibitory, antiplatelet, anti-inflammatory, and antioxidant activity [30,73,74].

In MMs, bioactive compounds such as triterpenes, sterols, phenolic compounds, and polysaccharide fractions possess metabolic-modulating capabilities [54]. These include blood pressure, glycaemia, cholesterol, triglyceride, and weight-lowering activities. The ACE inhibitory activity of several mushroom species was assessed via hot water and alcohol extracts [30]. *G. lucidum*, particularly as a hot water extract, and *Pleurotus* spp. demonstrated potent ACE inhibitory activity, which is assumed to be due to the phenolic content and antioxidant capacity. However, variations existed between species and depended on the extraction method [30]. In vitro digestion of *P. ostreatus* identified several peptides known to be ACE inhibitors [75]. A randomised, double-blind prevention trial is underway in the Democratic Republic of the Congo involving Tomeka^®^, a herbal mixture containing *A. bisporus* and other food-based nutrients such as soy, which is regarded for its potent ACE2 inhibition. The study aims to assess the intervention effect on COVID-19 markers of the RAAS, such as angiotensin-II and angiotensin-(1-7) [71]. Nutritional elements may support ACE inhibition indirectly by intercepting viral entry or via regulation and improvements in biomarkers associated with the involvement of the various systems [71]. For example, excessive sodium ions can impair the endothelial vasculature and risk hypertension, but manifestations may be ameliorated with higher potassium ion levels. Hence, mushrooms, which generally contain high potassium and low sodium may be a good nutritional source for ACE inhibition as well [1,24].

## 7. Comorbidities and Mortality Risk Reduction or Mitigation

### 7.1. Cardiometabolic Disorders Associated with COVID-19

Compounds of MMs have demonstrated biological activity with the potential to reduce the risk of cardiometabolic disease and comorbidity effects associated with COVID-19. One-third of patients with COVID-19, aged 40–60 years, have been identified as being afflicted with comorbidities such as CVD and hypertension [7]. Metabolic disorders, including obesity, diabetes, and hyperlipidaemia, are also featured in disease severity [7,8]. Additionally, meta-analyses have identified increased mortality risk and the need for intensive care for older patients with cardiovascular morbidities [7].

Fruiting bodies and mycelium extracts of edible mushrooms (some more than others) can be a valuable source of lovastatin, which is a statin group compound [76]. This compound inhibits HMG-CoA reductase, which is the rate-limiting step in cholesterol biosynthesis. Thus, MMs with activity associated with lovastatin’s mechanisms of action may be a promising source of anti-hypercholesterolaemic agents. Most notably, such cholesterol-lowering potential has been observed in the fruiting bodies of *Pleurotus* spp. and others, including *A. bisporus* and *H. erinaceus* [66,69,75,76,77]. This potential, along with the absorption and stimulatory effect of dietary fibres and gut effects on faecal excretion, makes for a promising functional food application.

The effects of various MMs have also been studied with other cardiometabolic parameters. Dicks and Ellinger [75] undertook a systematic review of eight clinical studies of subjects with and without type 2 diabetes mellitus (T2DM) using fresh, cooked, or dry powder *P. ostreatus*. These studies seemed to reveal beneficial effects, although the risk of bias was high or unclear due to methodological weaknesses and/or inadequate reporting in most studies. Nevertheless, observed were effects in glycaemic control (reduction in fasting and/or 2 h postprandial glucose), lipids metabolism (decrease in total cholesterol (TC), LDL cholesterol, and/or triglycerides), some reduction in blood pressure, antioxidant effects, and a decrease in food intake with no weight change [75]. The administration of total polysaccharides extracted from *P. ostreatus* was given for four weeks in a high-fat diet and streptozotocin (STZ)-induced type 2 diabetic rats [78]. Elevated blood glucose levels were reduced, insulin resistance improved, and glycogen increased. The mechanisms occurred through the activation of GSK-2 phosphorylation in the liver and GLUT4 translocation in muscle tissue. In high-fat-fed rats, *A. bisporus* demonstrated increased high-density lipoprotein (HDL) along with reductions in TC and LDL, and additionally, in type 2 diabetic rats induced by STZ, glucose levels decreased [79]. The anti-hyperglycaemic effect, along with antioxidant protection on the pancreas, kidney, and liver, were also demonstrated with *H. erinaceus* polysaccharides on STZ-induced rats [80].

The pharmacological effects of *C. militaris* SU-12 residue polysaccharide were investigated in a study utilising high-fat emulsion-induced hyperlipidaemic mice models. Identified were the characteristics of the polysaccharides, concluding these to have demonstrable antihyperlipidaemic, hepatoprotective activities, and an increase in antioxidant activity when serum and liver sections were analysed [81]. STZ-induced type 1 diabetic rats were used in a study that compared *G. frondosa* (water-soluble powdered whole Maitake fraction SX), two anti-diabetic drugs, and a control to assess circulating glucose levels and blood pressure (BP) [82]. All treatments generally decreased circulating glucose levels compared to control. However, only the Maitake group consistently demonstrated enhanced insulin sensitivity, significantly lowered systolic BP plus a decrease in the RAAS, and increased nitric oxide system activity. *F. velutipes* powder and extract were shown to have good antioxidant activity. They were a rich source of dietary fibre and mycosterol capable of impacting and reducing cardiometabolic disease parameters such TC, LDL cholesterol, and triglycerides (TGs) [83]. However, *G. lucidum* has produced less consistent results. In a study with high-fat-fed rabbits, Li et al. [84] assessed that atherosclerotic plaques were attenuated along with a reduced generation of ROS and malondialdehyde.

In contrast, no positive results were revealed in a prospective double-blind randomised, placebo-controlled trial of 84 subjects with type 2 diabetes mellitus and metabolic syndrome over 16 weeks using *G. lucidum*, *G. lucidum* with *C. sinesis*, or a placebo [85]. The primary outcome measures were blood glucose biomarkers (glycosylated haemoglobin (HBA1c) and fasting plasma glucose), and secondary outcome measures were HDL and LDL, TGs, BP, C-reactive protein, and apolipoproteins A and B markers. When the two intervention groups were combined due to sample size inadequacy, there was no effect on either the primary or secondary outcomes [85]. In a secondary analysis to investigate the benefits of daily intake of *A. bisporus* on cardiometabolic risk, the stored serum of prediabetic patients with features of metabolic syndrome were analysed. No significant changes to body weight, cardiovascular or metabolic parameters were observed, and plasma leptin did not change. Nevertheless, ergothioneine concentrations, oxygen radical absorbance capacity, and adiponectin were increased, with a reduction of advanced glycation end products (AGEs) [86].

### 7.2. Gut Microbiota Modulation as a COVID-19 Risk Reduction Consideration

It is recognised that the intestinal flora structure, the abundance and diversity of microbiota, or activity can influence positive or negative health outcomes. For example, aberrances of gut microbiota may contribute to chronic inflammatory diseases such as atherosclerosis, thrombosis, diabetes, and asthma, which may be in part due to oxidative stress [87]. On the other hand, symbiosis reduces cardiovascular and other metabolic diseases’ risks, lowers postprandial blood glucose, increases satiety for weight management, and improves laxation [88]. These are all factors associated with COVID-19 disease severity and are also lifestyle associated; therefore, they are modifiable and applicable to lowering risk. Fortunately, the functions and diversity of gut microbiota are influenced by the non-digestible and digestible fibres and prebiotic properties of mushrooms. Ma et al. [25] reviewed collated studies from various disease models that demonstrated health improvement from EMPs. Correlations between EMPs and beneficial host microbiota suggest regulatory effects [25]. Similar effects may also be induced via mannitol, raffinose, resistant starch, and chitin found in mushrooms [88]. Benefits of EMPs highlighted by Ma et al. were metabolic improvements such as reduction in TC and other lipid markers, anti-obesity activities, inflammation and insulin resistance, improved gut mucosa integrity and intestinal morphology, signalling pathways, and pro-inflammatory cytokines inhibition [25].

Short-chain fatty acids (SCFAs), including acetic acid, propionic acid, butyric acid, and valeric acid, are known to provide beneficial health effects such as nutrient supply to the colonic epithelium, oxidative stress reduction, immune stimulation, and colonic pH modulation [87]. However, the proportion of SCFAs varies depending on the EMPs from different mushrooms [25]. *G. lucidum* appears most significant for microbiota-derived health effects. Specifically, *G. lucidum* polysaccharide (GLP) appeared to have notable anti-diabetic and anti-obesity effects [89,90]. For example, GLP restored the gut microbiota of T2DM rats to a normal level and modified metabolites [91], and GLPS3 (*G. lucidum* mycelium polysaccharide strain S3) increased the relative abundance of beneficial bacteria Lactobacillus, Roseburia, and Lachnospiracea in mice induced with repetitively intraperitoneal injection of diethyldithiocarbamate [87]. An increase in Roseburia after dietary administration with GLPS3, was considered an indicator of the health-promoting activities of the SCFAs and possibly also of immune stimulation.

Contrasting findings were reported by Lee et al. [92] when examining results from the Nurses’ Health Study (for women) and the Health Professionals Follow Up Study (for men). In these studies, respondents were asked how often they consumed fresh, cooked, or canned mushrooms (species unidentified). Data collected were short and long-term changes in biomarkers such as lipids, insulin, inflammation, or cardiovascular risks such as sex, lifestyle factors, and certain medical conditions. No benefit of mushroom consumption on cardiovascular disease and T2DM was elucidated. However, a potential inverse relationship to T2DM was suggested.

Table 1 summarises the research findings exhibiting pro-health effects and benefits resulting from bioactive compounds or secondary metabolites of selected MMs, which are deemed applicable for potential COVID-19 therapy consideration.

## 8. Considering Limitations or Barriers in the Application of Mushrooms as a Medicine

Many factors may impact nutrient characteristics and potentially optimal and efficacious bioactivity of any particular plant compound, including MMs. These involve environmental influences such as plant provenance (e.g., geographic region, climate, and temperature); cultivation practices and production methods; plant structure; and isolation and extraction techniques and methods [4,29,55,70,93,94]. For example, in a study by Cohen et al. [95], fifteen dried and crushed Basidiomycetes MM strains, fruiting bodies, and mycelia were analysed. The protein and carbohydrate content ranges were 8.6–42.5% and 42.9–83.6%, respectively. Varying results for macro and micronutrients, and of concern, possible toxic elements, were reported as well. These results seem consistent with previous findings [24]. Therefore, consuming mushrooms with the intention of deriving some health or medicinal benefit may need to consider these influences to enhance or mitigate inhibitory effects.

The extraction methods and the part of the plant utilised appear to significantly impact bioactive potential. As an example, Pop et al. [96] demonstrated the isolation of a range of biocompounds from *Trametes* spp. using three different extraction solvents of water, ethanol, and methanol. Even within the same species, the bioactivity might differ depending on the type of extraction (water or ethanol/methanol) and the structure/section of the fruiting body [5,29,55,69].

Friedman’s review [29] on *H. erinaceus* isolated specific bioactive secondary metabolites using various solvents and according to the different plant structures such as mycelia and the polysaccharide fractions. In cultivated *H. erinaceus* mycelia, ergosterol content and new sterols were isolated. Antioxidant properties in a water-soluble polysaccharide were identified but not in an ethanol extraction of another polysaccharide within the same species. Ma et al. [25] comprehensively examined the role EMPs have in their activities against obesity, inflammatory bowel disease, and cancer. They characterised the type of polysaccharide along with the extraction method and demonstrated that the carbohydrate structure and chemistry, and thus the methods to isolate, are vital information for assigning effects. Martel et al. [55] surveyed studies on cells, animals, and humans utilising various constituents of mushrooms and plants. Depending on the extraction method, the effects on a wide variety of immune cell types were either stimulatory or inhibitory. The dichotomy appeared to be due to the differential solubility and potency of the main constituents in the extracts. Within the same species, water-soluble polysaccharide extracts activated immune responses, but ethanol extracts inhibited them, although there were a few exceptions.

It is recognised that improved techniques and technologies may remedy disadvantages associated with the extraction of bioactive compounds, such as long extraction times, low selectivity, and solubility [53,97]. As an example of the potential associated with improved technologies, Wu et al. [53] undertook a study to evaluate the anti-inflammatory effects of total polysaccharides and β-glucans extracted from *G. frondosa* mycelia. The potential mechanisms were evaluated by examining effects on nitrite, prostaglandins, pro-inflammatory cytokines (TNF-α, IL-6 and IL-1β), and intracellular reactive oxygen species in lipopolysaccharide-induced macrophage cells. They compared a conventional extract method using ethanol with three different high pressure-assisted extractions (PE-200, PE-400, PE-600). The PE method in each case yielded greater extraction and content of polysaccharides and β-glucans and exerted stronger anti-inflammatory activities than the conventional method. Benson et al. [26] demonstrated that three components of *T. versicolor* (mycelium, initial substratem and fermented rice flour substrate), plus different extractions and combinations, could produce varied immune responses and activities when tested on human peripheral blood mononuclear cell cultures.

These highlights demonstrate that there may be variations in the isolates depending on the methods utilised. Additionally, more modern and improved techniques may produce a superior raw product, potentially creating differences in the efficacies associated with mechanisms of action and health benefit claims. Therefore, it seems essential to their application to expect mushroom therapeutic treatments to be produced from regions and in conditions that result in the most efficacious response, according to the most current research. As more and more commercialised products are becoming available for general purchase, as well as in therapeutic settings, it is vital to ascertain pharmacokinetics, timing of administration, dosage, mode, and formulation variances [27]. These are all factors to be considered in the application of bioactive compounds. This can only be accurate and informed if the whole plant or extract characteristics have been clearly identified and elucidated. Figure 2 summarises important considerations and some potential limitations as presented in this appraisal. Key inputs into isolating bioactive compounds are influenced by many factors. Extraction methods, in particular, appear to be very important. Effects from the extracts from the same and different species can vary and even produce an opposite effect in activity. Beyond this, other factors need consideration if MMs are to be utilised for therapeutic and treatment purposes.

## 9. Conclusions

This review identified major findings of in vitro, in vivo, and human studies for mushrooms utilised in the medicinal context. Particular species were used as examples and examined for their health effects due to their ready availability and/or use in nutraceutical products. COVID-19 pathogenesis presents a spectrum of systemic health impacts for consideration. In particular, this relates to the aberrant immune and inflammatory responses that can lead to severe and detrimental consequences. It was demonstrated that specific bioactive compounds derived from mushrooms are capable of inducing a physiological effect that could be considered applicable in preventing COVID-19, mitigating symptoms or reducing disease severity. These effects include anti-pathogenic, anti-inflammatory, immune-modulatory, antioxidant, and ACE inhibitory activity. Thus, MMs are well placed as a possible therapeutic option due to these functions and properties.

Increasingly, MMs are being commercialised and promoted for their health-promoting benefits. However, potentially, many of the health claims may be ahead of the actual research validation, particularly concerning human translatability in specific contexts—in this case, COVID-19. Many factors such as extraction method, growing conditions, standardisation practices, mechanisms of action, and formulation synergies require greater understanding and examination. Hence, continued efforts must be applied to realising more research and improving data collection and modelling, particularly in humans, with all these variables and factors in mind.

## Figures and Tables

**Figure 1 molecules-27-02302-f001:**
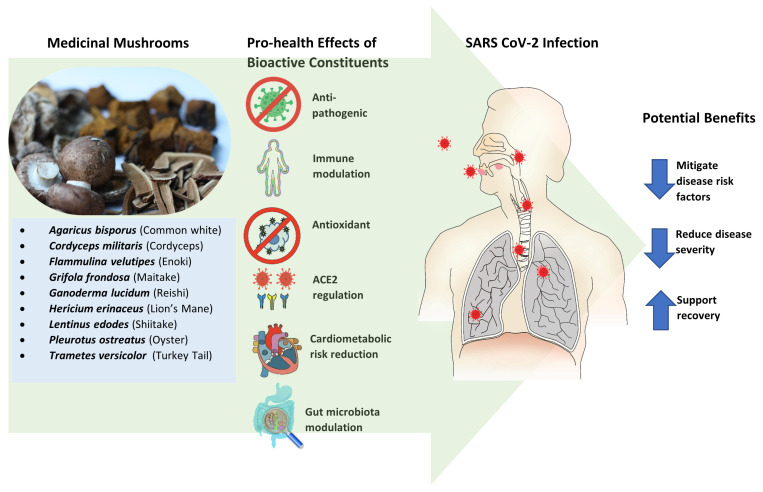
Medicinal mushrooms demonstrated modulatory and regulatory effects via the actions of bioactive compounds. These effects may apply in the pathophysiology and sequelae of SARS CoV-2 infection in humans. Potential benefits to consider and investigate specifically in this context involve mitigating disease risk factors, reducing disease severity, and supporting recovery.

**Figure 2 molecules-27-02302-f002:**
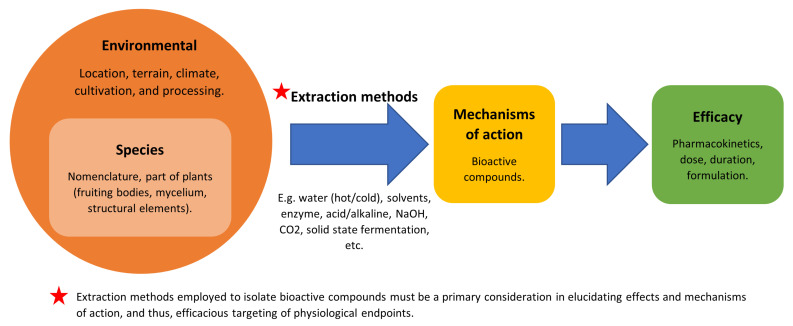
Factors to consider if medicinal mushrooms are to be utilised for therapeutic and treatment purposes.

**Table 1 molecules-27-02302-t001:** Research findings for specific mushroom species demonstrating beneficial effects and activities for potential pro-health therapeutic application.

Mushroom	Bioactive Compound	Pro-Health Effects	References
*A. bisporus* (Common white)	Phenolics e.g., Ergothioneine	Antioxidant; increased ORAC activity; increased adiponectin; reduced AGEs; increased glutathione reductase and catalase activities	[63,70,86]
	Polysaccharides	Gut microbiota regulation; intestinal barrier integrity	[25]
	Secondary metabolites	Anti-hyperglycaemic; inhibitory effects on LDL oxidation; reduced HMG-CoA reductase activity; LPS reduction; ACE2 inhibition; cardiometabolic parameters improvement	[30,66,79,80]
Andosan™	Ergosterol	Cytotoxic	[51]
	Commercial extract AbM + *H. erinaceus* and *G. frondosa*	Immunomodulatory; anti-inflammatory; anti-tumour	[49]
Tomeka™	Commercial extract *A. bisporus* + soy	Anti-pathogenic	[3,29]
*C. militaris* (Cordyceps)	Cordycepin	Antiviral; RNA synthesis inhibition; suppressed EB viral replication; cytotoxic	[14,41]
		Antioxidant; anti-hyperlipidaemic; hepatoprotective	[81]
	SCFAs	Immune regulation and health promoting	[25]
*F. velutipes* (Enoki)	Polysaccharides/dietary fibre	Reduced cardiometabolic parameters	[83]
	Antioxidant	Mycosterol	[83]
*G. frondosa* (Maitake)	Polysaccharides	Anti-inflammatory; antioxidant; immunomodulatory	[31,53]
		Increased insulin sensitivity; decreased systolic BP; decreased RAAS; increased NO	[82]
*G. lucidum* (Reishi)	Polysaccharides	SCFAs production; gut microbiota regulation; anti-obesity; anti-inflammation; reduced metabolic endotoxaemia; decreased FBG and insulin levels	[87,89,90,91]
	Ganodermic compounds—triterpenoids, other phenolics	Antioxidant; atherosclerotic plaque attenuation; anti-tumour; anti-inflammation; antiviral HIV-1 and HIV-1 protease inhibition	[30,42,56,64,84]
	Mycelia fractions	ACE inhibition	[73]
*H. erinaceus* (Lion’s Mane)	Mycelia polysaccharide fractions	Anti-hyperglycaemic; improved antioxidant enzymatic activities	[69,80]
	Fruiting body solvent fractions	LDL oxidation inhibition; HMG-CoA reductase inhibition	
*L. edodes* (Shiitake)	Polysaccharides; β-glucans	Antiviral; antioxidant; immunomodulatory; cytotoxic; anti-inflammatory; microbiome regulation	[1,2,6,17,25,44]
	Aqueous extract	Anti-bacterial; anti-fungal	[43]
	Lovastatin	Hypolipidaemic	[76]
*P. ostreatus* (Oyster)	β-glucans-pleuran	Immunomodulatory	[36,46,47,48]
	Lectins	Vaccine adjuvant	[20,45]
	Phenolics, peptides	ACE2 inhibition	[30,75]
	Lovastatin	HMG-CoA reductase inhibition	[75,76,77]
	Polysaccharides	Cardiometabolic parameter improvements	[75,78]
*T. versicolor* (Turkey Tail)	PSP, PSK	Cancer therapy/adjuvants; Immunomodulatory	[57,59,60,61]
		Gut microbiota modulation	[25]
	Glucans, phenolics	Antioxidant	[62]

## Data Availability

Not applicable.

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
