# Peer review of "Health-Promoting Properties of Medicinal Mushrooms and Their Bioactive Compounds for the COVID-19 Era—An Appraisal: Do the Pro-Health Claims Measure Up?"

_molecules, 2022, doi:10.3390/molecules27072302_

Round 1

Reviewer 1 Report

This is a very interesting review article giving a significant contribution to the better understanding of a current topic of usage of natural products in the treatment of COVID-19. The article is well organized and written, giving a comprehensive review of the relevant literature, and appropriate interpretation of the available data. Therefore it is my opinion that it is suitable for publication in the present form.

Author Response

Thank you for the positive appraisal. We appreciate the time and effort made in reviewing our manuscript.

Reviewer 2 Report

Comments are included in the appendix 

Author Response

We truly appreciate the reviewer’s feedback which helps to improve the manuscript considerably. Please see the attached file for our point-by-point responses to the comments.

Reviewer 3 Report

This MS deals with the question of suitability of medicinal mushrooms and their extracts to be used as for COVID-19 cure.

This is important topic that raises important questions related to the use of natural biological resources in medicine, as the produces might suffer from batch -related differences in the composition of each products. This is due to effect of environmental conditions at the growing time, biogeography, extraction method and more, as mentioned by the authors. The idea of the use of medicinal mushrooms to combat COVID-19 is not novel, nor the obstacles raised in using medicinal plants for commercial product to cop with health problems is novel. Yet, the combination of reviewing the medicinal mushrooms properties and the list of subjects need attentions to be paid to obtain repeated, clear and unified product is important. It gives an overview from both benefits of using MM as well as the limitations to overcome. All which are important to wide range of readers.

Comments:

  1. I am very concerned that almost identical paper of the same authors, with similar title is already available online, although in "preprint mode" . This makes the official publishing the MS in this journal unnecessary.  
  2. There are few editing issues:

Line 584- Ref 8: Please erase the parentheses around the article's title.

Figure 1, 2: Titles should give explanation of what we see in the figure and not results and discussion. This should be in the text.  In Figure 2 there is a title on top of the figure itself. My suggestion is to move this title to the figure title below.

Author Response

We thank the reviewer for the valuable feedback. We have fixed the editing issues. Specifically:

Line 584- Ref 8: 

Please erase the parentheses around the article’s title.

Done

(Line 607 in the updated manuscript – clean version)

Figure 1, 2:

Titles should give explanation of what we see in the figure and not results and discussion. This should be in the text. In Figure 2 there is a title on top of the figure itself. My suggestion is to move this title to the figure title below.

We have updated Figure 2 as per your suggestion. The original legend is now part of the main text leading to Figure 2. (See line 550-554 of the updated manuscript - clean version).

We also acknowledged the reviewer’s concern regarding the existence of a manuscript preprint available online. However, we wish to elaborate that a preprint is merely a draft version, and its availability online does not equate to journal publication. At submission, we have informed the journal editorial office of the preprint and confirmed that the manuscript was not currently under consideration by other journals. 

It is stated on Molecules’ Author Guidelines that “Molecules accepts submissions that have previously been made available as preprints provided that they have not undergone peer review.”

In fact, MDPI, the publisher of Molecules, operates a Preprint site at https://www.preprints.org/ and encourages authors submitting to MDPI journals to deposit their manuscripts on this preprint site.

Thank you very much. 

Round 2

Reviewer 2 Report

I accept the corrections and clarifications made by the authors.

Reviewer 3 Report

The paper is much improved. I like the positive words pro-health instead of Anti. 

Please go over the English editing for few small errors.

I have no from the authors reply to my comment regarding the already published  your similar  paper in the internet.